# Predictive Effects of FT3/FT4 on Diabetic Kidney Disease: An Exploratory Study on Hospitalized Euthyroid Patients with T2DM in China

**DOI:** 10.3390/biomedicines11082211

**Published:** 2023-08-07

**Authors:** Xin Zhao, Jianbin Sun, Sixu Xin, Xiaomei Zhang

**Affiliations:** Department of Endocrinology, Peking University International Hospital, Beijing 102206, China; zhaoxin1@pkuih.edu.cn (X.Z.);

**Keywords:** type 2 diabetes mellitus, thyroid hormone, glycosylated hemoglobin, diabetic kidney disease, inflammation

## Abstract

Objective: This study aims to explore the correlation between the free-triiodothyronine (FT3)-to-free-thyroxine (FT4) ratio (FT3/FT4) and diabetic kidney disease (DKD) in patients with type 2 diabetes mellitus (T2DM). Methods: This study retrospectively analyzed 1729 patients with T2DM hospitalized in the Department of Endocrinology, Peking University International Hospital, from January 2017 to August 2021, including 1075 males and 654 females. In accordance with the FT3/FT4, the patients were divided into three groups. Results: (1) The levels of glycosylated hemoglobin (HbA1c), fasting blood glucose (FBG) and postprandial blood glucose (PBG) among the three groups were significantly different, with the low FT3/FT4 group having the highest HbA1c, FBG and PBG among the three groups (F = 39.39, *p* < 0.01; F = 27.04, *p* < 0.01; F = 5.76, *p* = 0.03; respectively). (2) The proportion of DKD is the highest in the low FT3/FT4 group and the lowest in the high FT3/FT4 group (χ^2^ = 25.83, *p* < 0.01). (3) Logistic regression showed that low FT3/FT4 were independent risk factors for DKD (OR = 2.36, 95 CI% 1.63, 3.43; *p* = 0.01). Conclusion: A decrease in the FT3/FT4 is an independent predictor of DKD occurrence in patients with T2DM.

## 1. Introduction

In recent times, due to the advancement of the economy and people’s lifestyles, there has been a gradual rise in the prevalence of type 2 diabetes mellitus (T2DM). According to the International Diabetes Federation, there are approximately 422 million people with DM worldwide [1]. End-stage renal disease (ESRD) is one of the most common causes of end-stage diabetic kidney disease (DKD), making it a significant financial burden for patients and greatly affecting their long-term quality of life. DKD is not only a heavy economic burden for patients but also has a great impact on their long-term quality of life. Ultrafiltration is a common feature of early DKD. It is possible for individuals with DKD to experience a significant decline in eGFR, even if their initial eGFR levels are within the normal range [2]. In the early stages of DKD, individuals may not experience any obvious clinical symptoms, and both albuminuria and eGFR levels may remain within the normal range. This can lead to missed opportunities for early diagnosis and intervention. The active exploration of the risk factors of DKD can provide a sufficient clinical basis for the prevention and treatment of DKD and the improvement of outcomes in patients with T2DM.

Earlier research has established a strong association between thyroid hormone (TH) and DM, as well as its associated complications. The association between thyroid dysfunction and microvascular complications in diabetic patients can be attributed to multiple factors including metabolic disorders, acid–base imbalances and tissue hypoxia caused by hyperglycemia, which can inhibit the activity of type 1 deiodinase and reduce the conversion of T4 to T3 in peripheral tissues [3]. It has been reported that the FT3 level is positively correlated with islet β cell function in patients with pre-diabetes [4]. Many studies have shown a close correlation between THs and insulin resistance [5,6]. Postmenopausal status with metabolic syndrome is associated with increased TSH, FT3 and FT4 levels and insulin resistance in obese women [7,8]. Also, a decrease in TH can directly reduce ventricular diastolic function and cardiac output, increase peripheral vascular resistance, reduce renal blood flow and ultimately affect renal function.

Compared with healthy people, the prevalence of thyroid disease in DM patients is higher; in particular, the prevalence of hypothyroidism in DM patients is higher than in healthy people [9]. Recent studies showed that there is a correlation between reduced levels of free triiodothyronine (FT3)/free thyroxine (FT4) and an elevated inflammatory burden [10]. Currently, the majority of studies investigating the link between thyroid hormone and DKD focus on patients with overt hypothyroidism or subclinical hypothyroidism (SCH). There are limited studies examining the relationship between TH and DKD in patients with T2DM who have normal thyroid hormone levels. Some researchers found that FT3 in patients with T2DM who had normal TH levels was negatively correlated with DKD [11]. However, the study had a restricted sample size and did not establish a causal relationship between TH and DKD. In patients with T2DM with normal TH levels, exploring the relationship between TH and DKD holds great clinical significance in predicting the risk of developing DKD and guiding preventative measures. Therefore, this study aims to explore the correlation between TH and DKD in patients with T2DM with normal TH levels to provide a clinical basis for discussing the impact of TH levels on DKD in T2DM patients.

## 2. Materials and Methods

### 2.1. Ethics Statement

The study was approved by the Ethics Committee of Peking University International Hospital and was conducted in accordance with the ethics standards of institutional and national research committees and the 1964 Helsinki Declaration and its later amendments or comparable ethics standards. As the study involved the retrospective analysis of clinical data, the requirement for written informed consent was waived. The number of the ethics approval is 2022-KY-0030-01. The clinical trial register number is ChiTR1800014785.

### 2.2. Research Subjects

The present study conducted a retrospective analysis of a total of 1729 patients diagnosed with T2DM who were admitted to the Department of Endocrinology at Peking University International Hospital from January 2017 to August 2021. The study population comprised 1075 males and 654 females, with an average age of 54.67 ± 13.65 years. All participants in this study were diagnosed with diabetes according to the diagnostic criteria for diabetes established by the World Health Organization (WHO) in 1999 [12]. The criteria include (1) experiencing typical symptoms of diabetes with random blood glucose levels ≥ 11.1 mmol/L; (2) fasting blood glucose levels ≥ 7.0 mmol/L; and (3) blood glucose levels ≥ 11.1 mmol/L for 2 h after ingesting 75 g of glucose during oral glucose tolerance tests. Patients who did not exhibit any symptoms of diabetes were required to undergo repeated testing on another day. The exclusion criteria for this study were as follows: (1) patients with type 1 diabetes, gestational diabetes and other special types of diabetes; (2) patients with acute complications of diabetes; (3) patients with urinary tract infection, hematuria (including the menstrual period) and non-DKD or ESRD requiring dialysis; (4) patients with a history of thyroid disease (such as thyroid dysfunction or thyroidectomy), those taking thyroid drugs (such as levothyroxine or antithyroid drugs) or those with dysfunction in the parathyroid gland, adrenal gland or pituitary gland; and (5) patients with hematological diseases and malignant tumors.

### 2.3. Methods

#### General Conditions and Laboratory Biochemical Indices

A retrospective analysis was conducted on the data of 1729 patients, and their general characteristics such as sex, age, height, weight, waist circumference, hip circumference, systolic blood pressure (SBP), diastolic blood pressure (DBP), diabetic duration and complications of diabetes were recorded in detail. BMI and WHR were calculated using the following formulae: BMI (kg/m^2^) = weight (kg)/body height^2^ (m^2^); WHR = waist circumference (cm)/hip circumference (cm).

All participants were required to fast for more than 8 h, and venous blood was drawn on the following morning. The biochemical parameters that were measured included glycosylated hemoglobin (HbA1c), fasting blood glucose (FBG), postprandial blood glucose (PBG), triglyceride (TG), total cholesterol (TC), high-density lipoprotein cholesterol (HDL-C), low-density lipoprotein cholesterol (LDL-C), serum creatinine (sCr), uric acid (UA), free thyroxine (FT4), free triiodothyronine (FT3), thyroid-stimulating hormone (TSH) and the ratio of FT3 to FT4 (FT3/FT4). The PBG were tested for 2 h after ingesting 75 g of glucose during oral glucose tolerance tests. The participants were asked to provide urine samples for three consecutive mornings. The samples were used to measure urinary microalbuminuria, urinary creatinine and urinary-albuminuria-to-creatinine ratio (UACR) using the immunoturbidimetric method in our laboratory.

Glomerular filtration rates (eGFRs) were calculated based on the sCr of the study subjects. Calculation formulae are as follows:

Male:sCr ≤ 0.9 mg/dL: eGFR CKD-EPI-ASIA = 141 × (sCr/0.9) − 0.411 × 0.993 age × 1.057
sCr > 0.9 mg/dL: eGFR CKD-EPI-ASIA = 141 × (sCr/0.9) − 1.209 × 0.993 age × 1.057

Female:sCr ≤ 0.7 mg/dL: eGFR CKD-EPI-ASIA = 141 × (sCr/0.7) − 0.329 × 0.993 age × 1.049
sCr > 0.7 mg/dL: eGFR CKD-EPI-ASIA = 141 × (sCr/0.7) − 1.209 × 0.993 age × 1.049

According to the Chinese Guidelines for the prevention and treatment of diabetes kidney disease [13], DKD can be diagnosed in patients with at least one of the following conditions when DM is identified as the cause of kidney damage and other causes of CKD are excluded: (1) UACR ≥ 30 mg/g in at least two out of three tests, after excluding any interfering factors; (2) eGFR < 60 mL·min^−1^·(1.73 m^2^)^−1^ for more than 3 months; or (3) renal biopsy consistent with the pathological changes in DKD.

In this study, a total of 384 patients with T2DM met the criteria for a DKD diagnosis. Out of these patients, 369 underwent renal biopsy, resulting in a renal biopsy rate of 96.09%. The others met the criteria: UACR ≥ 30 mg/g in at least two out of three tests, after excluding any interfering factors.

The patients in the study were categorized into three groups based on their FT3/FT4 levels. These groups were identified as the low group (FT3/FT4 < 0.25): *n* = 575 cases; the medium group (FT3/FT4 0.25–0.29): *n* = 577 cases; and the high group (FT3/FT4 > 0.29): *n* = 577 cases.

### 2.4. Statistical Analysis

The statistical analysis was performed using the SPSS Version 22.0 software (IBM, Chicago, IL, USA). Normality analysis was performed using the Kolmogorov–Smirnov test. The variables that followed normal distribution were presented as mean ± standard deviation. The comparison of continuous variables among the three groups was done using one-way analysis of variance (ANOVA), while the comparison between the two groups was done using the least-significant-difference (LSD) method. The comparison of counting data was done using the χ^2^ test. Unconditional logistic regression models were used for univariate and multivariate analysis of the factors, and OR and its 95% CI were calculated. The receiver-operating-characteristic curves (ROC) were plotted and the area under the ROC curve (AUC) was calculated. All statistical tests were two-sided, and *p* < 0.05 was considered statistically significant.

## 3. Results

### 3.1. Comparison of General Characteristics and Biochemical Indexes among the Three Groups

The low FT3/FT4 group had the highest average age and the lowest BMI compared to the other two groups (*p* < 0.01). Additionally, the proportion of male patients was the highest in the high FT3/FT4 group (*p* < 0.01). The low FT3/FT4 group had the longest diabetic duration, while the high FT3/FT4 group had the shortest diabetic duration (*p* < 0.01). The UACR and eGFR levels among the three groups were significantly different, with the low FT3/FT4 group having the highest UACR and lowest eGFR levels (*p* = 0.02). The HbA1c, FBG and PBG levels among the three groups were significantly different, with the low FT3/FT4 group having the highest HbA1c, FBG and PBG levels (*p* < 0.05, respectively). In addition, the proportion of DKD was the highest in the low FT3/FT4 group and the lowest in the high FT3/FT4 group (*p* < 0.01). The SBP, DBP, UA, TC, TG, LDL-C and HDL-C levels were not significantly different among the three groups (*p* > 0.05) (Table 1).

### 3.2. Correlation Analysis between Thyroid Hormone and Renal Function (UACR, eGFR)

Correlation analysis revealed a negative correlation of UACR with FT4, FT3 and FT3/FT4 (r = −0.06, *p* = 0.03; r = −0.13, *p* < 0.01; r = −0.07, *p* < 0.01, respectively) and positive correlation of eGFR with FT4, FT3 and FT3/FT4 (r = 0.06, *p* = 0.01; r = 0.20, *p* < 0.01; r = 0.19, *p* < 0.01, respectively) (Table 2).

### 3.3. Logistic Regression Analyses of TH and DKD

Using DKD as the dependent variable and TH as the independent variable, a univariate logistic regression model was established. The results revealed that even after adjusting for factors such as age, BMI, sex, diabetic duration, blood pressur, and blood lipid and HbA1c levels, low levels of FT3/FT4 and low FT3 remained independent risk factors for DKD (Table 3).

### 3.4. Univariate Predictive Model of DKD with TH

The model to predict the risk of DKD using three variables, namely, FT3, FT4 and FT3/FT4, showed that the AUC of the models ranked FT3/FT4 (0.65) > FT3 (0.61) > FT4 (0.57). “Cut-off” is the value corresponding to the highest diagnostic accuracy of the variable for DKD. The corresponding cut-off points for FT4, FT3, and FT3/FT4 were 15.95 pmol/L, 4.85 pmol/L and 0.29 pmol/L, respectively (Table 4 and Figure 1).

### 3.5. Multivariate Predictive Model for the Risk of DKD

The multivariate predictive model was established with DKD as the dependent variable, and using indices of significant differences in univariate logistic analysis including sex, age, BMI, diabetic duration, blood pressure, HbA1c and FT3/FT4 as independent variables. The regression equation is logit (DKD) = −4.75 − 10.04 ×FT3/FT4 − 0.24 × Sex + 0.01 × Age + 0.02 × Diabetic duration + 0.02 × SBP + 0.07 × BMI + 0.18 × HbA1c. The AUC was 0.71 (95% CI 0.68, 0.74), the specificity was 56.75%, the sensitivity was 73.39% and the accuracy was 70.89%. The multivariate predictive model was established with DKD as the dependent variable, and using indices of significant differences in univariate logistic analysis including sex, age, BMI, diabetic duration, blood pressure, HbA1c and as independent variables. The regression equation is logit (DKD) = −8.02796 − 0.10217 × Sex + 0.01339 × Age + 0.02537 × Diabetic duration + 0.01901 × SBP + 0.05798 × BMI + 0.22655 × HbA1c. The AUC was 0.68 (95% CI 0.64, 0.71), the specificity was 72.07%, the sensitivity was 55.83% and the accuracy was 67.94% (Figure 2).

## 4. Discussion

DKD is a chronic condition resulting from diabetes and is considered one of the primary microvascular complications of the disease. It is estimated to occur in 20–40% of diabetic patients and is the leading cause of renal failure in this population. If DKD progresses to an advanced stage, it can cause permanent damage to renal function, leading to ESRD. This is a significant cause of mortality and cardiovascular events in patients with T2DM. Therefore, identifying the risk factors associated with DKD can provide valuable clinical information for preventing and treating the condition, ultimately leading to improved outcomes for individuals with T2DM.

The association between thyroid dysfunction and microvascular complications in diabetic patients can be attributed to multiple factors. These factors include the following: (1) metabolic disorders, acid–base imbalances and tissue hypoxia caused by hyperglycemia, which can inhibit the activity of type 1 deiodinase and reduce the conversion of T4 to T3 in peripheral tissues [3]. (2) T3 can increase insulin secretion through a variety of mechanisms, such as reducing apoptosis of islet β cells and promoting insulin gene transcription and expression [14,15]. It has been reported that the FT3 level is positively correlated with islet β cell function in patients with pre-diabetes [4]. (3) Endothelial dysfunction is a key factor in the pathogenesis of diabetic vascular complications (including DKD) [16]. T3 can modulate endothelial activity. (4) TH affects renal function both directly through its effects on the kidney and indirectly through its influence on other systems [17]. A decrease in TH can directly reduce ventricular diastolic function and cardiac output, increase peripheral vascular resistance, reduce renal blood flow and ultimately affect renal function.

In a retrospective study involving 2832 participants in China, after propensity-score-matching (PSM) analyses, the results indicate that high concentrations of serum FT3 were associated with the significantly reduced risk of having moderate-risk to very-high-risk DKD stages [18]. In our study, the results were similar; it is found that in patients with T2DM and normal TH levels, a low level of FT3 within the normal range was linked to an increased risk of developing DKD. The lower the FT3 level, the higher the risk of DKD. After adjusting for age, sex, diabetic duration, blood pressure and levels of blood glucose and blood lipid, a low level of FT3 remained an independent risk factor for DKD. A study conducted in Korea on individuals with normal TH levels demonstrated that low levels of FT3 within the normal range were associated with an increased risk of chronic kidney disease [19]. Additionally, a cross-sectional study carried out on elderly Chinese individuals found a significant negative correlation between FT3 within the normal range and urinary microalbumin [20]. These studies suggest that low levels of FT3 within the normal range can increase the risk of DKD in patients with T2DM. However, there are still some differences between our study and other studies, some studies included patients with abnormal thyroid function, so we may believe that some patients with end-stage renal disease themselves have low T3 syndrome, which affects the correlation between FT3 levels and DKD. In our study, our patients were T2DM patients with normal thyroid function, and the impact of low T3 or low T4 syndrome on the study results was excluded. This may exclude other confounding factors and better reveal the correlation between TH and DKD.

There are three types of deiodinase enzymes in peripheral tissues that regulate thyroid hormone metabolism: type 1, type 2 and type 3. Type 1 exists in the plasma membrane of the liver, kidney, thyroid and pituitary gland, and can deiodinize the inner and outer rings of T4 simultaneously; type 2 has the ability to deiodinize the outer ring to transform T4 into T3; and type 3 inactivates T4 and T3 via iodization of the internal ring [21]. In peripheral organs (such as the liver, muscle, brown adipose tissue, etc.), T4 is transformed into T3, which then plays a role in controlling metabolism and energy usage in these organs [22]. The levels of FT3/FT4 are used as an indicator of deiodinase activity, which can reveal how sensitive peripheral organs are to changes in FT4 levels. Although some studies have demonstrated that the ability of body tissues to respond to TH decreases with age, this perspective has not been widely acknowledged [23]. In situations of hunger and inflammation, the levels of T3 and T4 may decrease, while the levels of TSH and hypothalamic thyroid-stimulating hormone (TRH) remain normal or decrease, which reflects the abnormal negative feedback regulation of the hypothalamic pituitary thyroid axis [24]. A cross-sectional study showed that raised deiodinase activities in women with normal TH were linked to increased BMI, and higher deiodinase activity was significantly associated with higher glucose level. A previous study confirmed the positive association between FT3/FT4 and DM [25,26]. However, the study included a relatively small number of patients and was retrospective, and fails to reveal the predictive value of FT3/FT4 on the occurrence of DM. Another study showed that associations between FT3/FT4 values and FBG, and concluded that FT3/FT4 was an independent risk factor for the development of gestational diabetes mellitus (OR = 1.31, 95% CI 1.18, 1.46) [27]. A previous study has proved the FT3/FT4 would be a significant predictor of homeostasis model assessment of IR and insulin levels [28]. Since abnormal glucose metabolism leads to hypothalamic–pituitary–thyroid axis dysfunction, the significance of the FT3/FT4 ratio for microvascular complications in T2DM patients remains elusive. In this study, the subjects were divided into three groups according to the level of FT3/FT4. The results showed that the proportion of DKD is the highest in the low FT3/FT4 group and the lowest in the high FT3/FT4 group. After adjusting for age, sex, diabetic duration, blood pressure and levels of blood glucose and blood lipid, a low level of FT3/FT4 remained an independent risk factor for DKD. At the same time, we used ROC to evaluate the predictive effect of FT3/FT4 on DKD. Additionally, it was found that the AUC for FT3/FT4 in predicting DKD was larger than that for FT3 alone. This indicates that FT3/FT4 has a stronger predictive advantage for DKD compared to FT3 alone. In this study, we established two models using DKD as the dependent variable, and the results showed that FT3/FT4 as the independent variable entered the model pair, significantly improving the accuracy of the model.

In this study, it was found that there was no significant correlation between TSH and DKD in the univariate logistic regression analysis. This is different from previous studies, which have shown that the TSH levels in patients with DKD are significantly higher than in those without DKD in patients with T2DM [29]. The discrepancies observed in our study compared to previous ones might be due to different reasons. First, there could be variations in the diagnostic criteria used for DKD across studies. Second, earlier research suggested that metformin treatment can lower TSH levels in patients with T2DM [30,31]. The varying proportions and doses of metformin used in different studies for patients with T2DM could have affected the analysis of the relationship between TSH and DKD. Finally, in SCH patients, TSH can fluctuate significantly, and the subjects of this study are limited to patients with T2DM and normal thyroid function. Different research subjects could have contributed to the results and shown differences in TSH and DKD correlation results. Therefore, the relationship between normal TSH levels and DKD remains unclear, and future wide-ranging clinical cohort studies are needed to further investigate this correlation.

In addition, the results of this study showed that age, male gender, diabetic duration, HbA1c levels and high blood pressure were also risk factors for DKD, which is consistent with previous research [32,33,34]. Furthermore, the multivariate predictive model was established with DKD as the dependent variable and sex, age, BMI, diabetic duration, BP, HbA1c levels and FT3/FT4 as independent variables.

There are some limitations in our study. First, since it was a retrospective study, we lacked data on certain indicators; for example, some inflammatory factors and the status of deiodinase 2 were not detected. Future studies are needed to investigate whether regulation of thyroid function can improve the prognosis of DKD, and prospective studies are required to explore the underlying mechanism and pathway of the relationship between FT3/FT4 and DKD. Second, the subjects of this study were all inpatients, which may not fully represent the correlation between TH and DKD in outpatients with T2DM. Moreover, since the data were collected during hospitalization, the results of eGFR, TH and other indicators were tested only once, which may have resulted in some degree of error. Finally, maybe due to the limitation of the study sample size, the ROC results need to be further improved. Nevertheless, the current research on the relationship between FT3/FT4 and DKD is limited, and our findings are only a preliminary exploration. In future studies, we will further expand the sample size and conduct multicenter studies to improve the research results of this study.

## 5. Conclusions

In conclusion, this study found that patients with T2DM and normal thyroid function who had low levels of FT3/FT4 had a significantly higher incidence of DKD. After adjusting for age, sex, diabetic duration, blood pressure and levels of blood lipid and blood glucose, low levels of FT3 and FT3/FT4 remained independent risk factors for DKD. These findings may be useful in predicting DKD in patients with T2DM.

## Figures and Tables

**Figure 1 biomedicines-11-02211-f001:**
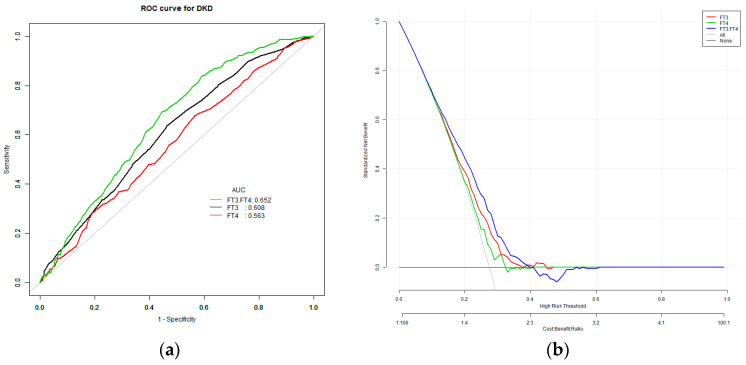
(**a**) The ROC curve for DKD. (**b**) The decision curve analysis for DKD. The overall predictive accuracy of FT3/FT4 for DKD was 0.65 (95% CI 0.62, 0.68); the overall predictive accuracy of FT4 for DKD was 0.57 (95% CI 0.54, 0.60); the overall predictive accuracy of FT3 of DKD was 0.61 (95% CI 0.58, 0.64).

**Figure 2 biomedicines-11-02211-f002:**
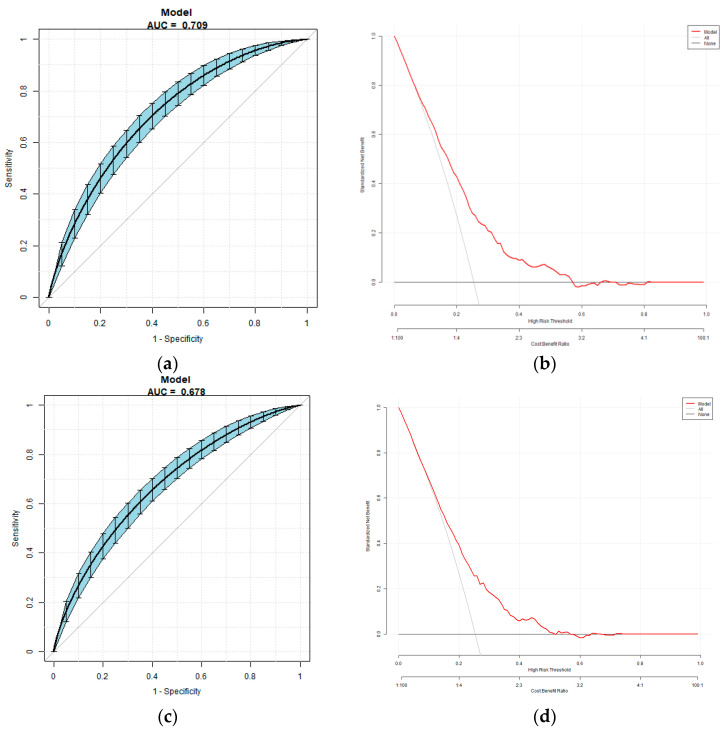
(**a**,**b**): The overall predictive accuracy of multivariate predictive model for the risk of DKD. (**c**,**d**): The overall predictive accuracy of multivariate predictive model for the risk of DKD excluding FT3/FT4 as independent variables.

**Table 1 biomedicines-11-02211-t001:** Comparison of general conditions and biochemical indexes among three groups.

Index	Low FT3/FT4	Medium FT3/FT4	High FT3/FT4	F (χ^2^)	*p*
	(*n* = 575)	(*n* = 577)	(*n* = 577)		
Age (years)	57.51 ± 13.83	54.53 ± 13.80 ^a^	51.96 ± 12.85 ^a,b^	24.32	<0.01
Sex (male%)	309 (53.74%)	354 (61.35%)	411 (71.23%)	41.06	<0.01
BMI (kg/m^2^)	25.35 ± 3.73	26.18 ± 3.80 ^a^	26.75 ± 3.86 ^a,b^	17.05	<0.01
Diabetic duration (years)	10.30 ± 9.22	9.50 ± 8.14	7.90 ± 6.83 ^a,b^	11.69	<0.01
SBP (mmHg)	132.60 ± 18.68	132.10 ± 17.96	132.02 ± 15.55	0.18	0.83
DBP (mmHg)	78.93 ± 11.07	79.05 ± 11.35	79.70 ± 10.82	0.81	0.44
WHR	0.94 ± 0.08	0.95 ± 0.07	0.96 ± 0.06 ^a^	5.13	0.02
TC (mmol/L)	4.35 ± 1.17	4.40 ± 1.12	4.40 ± 1.10	0.21	0.81
TG (mmol/L)	1.95 ± 1.62	2.06 ± 1.42	2.06 ± 1.79	1.39	0.20
LDL-C (mmol/L)	2.55 ± 0.93	2.62 ± 0.93	2.60 ± 0.90	0.74	0.48
HDL-C (mmol/L)	1.02 ± 0.28	1.01 ± 0.28	0.99 ± 0.29	2.07	0.13
UA (umol/L)	339.82 ± 98.78	348.50 ± 93.42	348.51 ± 84.59	1.57	0.21
eGFR	93.35 ± 22.30	98.95 ± 18.86 ^a^	101.63 ± 17.26 ^a^	24.73	<0.01
UACR (mg/g)	119.53 ± 440.90	61.08 ± 291.11 ^a^	57.96 ± 259.18 ^a^	5.39	0.02
HbA1c (%)	9.11 ± 2.12	8.38 ± 1.94 ^a^	8.02 ± 1.76 ^a,b^	39.39	<0.01
FBG (mmol/L)	9.41 ± 3.60	8.64 ± 3.06 ^a^	8.00 ± 2.58 ^a,b^	27.04	<0.01
PBG (mmol/L)	12.97 ± 4.74	12.62 ± 4.26	12.03 ± 4.03 ^a,b^	5.76	0.03
FT4 (pmol/L)	17.79 ± 1.85	16.58 ± 1.77 ^a^	15.22 ± 1.66 ^a,b^	306.65	<0.01
FT3 (pmol/L)	4.03 ± 0.46	4.51 ± 0.49 ^a^	4.86 ± 0.54 ^a,b^	412.15	<0.01
TSH (uIU/mL)	1.89 ± 0.88	1.97 ± 0.88	2.00 ± 0.89	2.66	0.07
DKD (%)	172 (29.9%)	118 (20.45%) ^a^	94 (16.29%) ^a,b^	25.83	<0.01

Note: ^a^: compared with the low FT3/FT4 group, the difference was statistically significant (*p* < 0.05). ^b^: compared with the medium FT3/FT4 group, the difference was statistically significant (*p* < 0.05). BMI is for body mass index, SBP is systolic blood pressure, DBP is for diastolic blood pressure, FBG is for fasting blood glucose, PBG is for postprandial blood glucose, HbA1c is for glycosylated hemoglobin, eGFR is for glomerular filtration rates, UA is for uric acid, UACR is for urinary-albuminuria-creatinine ratio, TC is for total cholesterol, TG is for triglycerides, LDL-C is for low-density lipoprotein cholesterol, HDL-C is for high-density lipoprotein cholesterol, FT4 is for free thyroxine, FT3 is for free triiodothyronine, TSH is for thyroid-stimulating hormone, DKD is for diabetic kidney disease.

**Table 2 biomedicines-11-02211-t002:** Correlation analysis between thyroid hormone and renal function (UACR, eGFR).

Index	UACR		eGFR	
	r	*p*	r	*p*
FT4	−0.06	0.03	0.06	0.01
FT3	−0.13	<0.01	0.20	<0.01
FT3/FT4	−0.07	<0.01	0.19	<0.01

Note: FT4 is for free thyroxine, FT3 is for free triiodothyronine, UACR is for urinary-albuminuria-creatinine ratio, eGFR is for glomerular filtration rates.

**Table 3 biomedicines-11-02211-t003:** Logistic regression analysis of FT3/FT4 and DKD.

Index	Crude OR	95% CI	*p*	Adjust OR	95% CI	*p*
FT3 (pmol/L)						
High (>4.68)	1.00			1.00		
Medium (4.20–4.68)	1.81	1.39, 2.36	0.02	1.74	1.21, 2.49	0.02
Low (<4.20)	2.20	1.68, 2.87	0.01	1.96	1.33, 2.90	0.03
FT4 (pmol/L)						
High (>17.39)	1.00			1.00		
Medium (15.66–17.39)	0.94	0.73, 1.20	0.61	0.93	0.67, 1.30	0.68
Low (<15.66)	0.71	0.55, 0.93	0.02	0.85	0.60, 1.19	0.34
FT3/FT4						
High (>0.29)	1.00			1.00		
Medium (0.25–0.29)	3.13	2.32, 4.22	<0.01	1.72	1.19, 2.49	0.02
Low (<0.25)	3.96	2.95, 5.32	<0.01	2.36	1.63, 3.42	0.01
TSH (uIU/mL)						
High (>2.29)	1.00			1.00		
Medium (1.45–2.29)	1.00	0.77, 1.30	0.99	1.02	0.73, 1.43	0.89
Low (<1.45)	0.95	0.74, 1.24	0.72	0.84	0.60, 1.17	0.30

Note: FT4 is for free thyroxine, FT3 is for free triiodothyronine, TSH is for thyroid stimulating hormone. Crude OR is for OR before adjusting age, BMI, sex, diabetic duration, blood pressure and blood lipid and HbA1c levels. Adjust OR is for adjusting age, BMI, sex, diabetic duration, blood pressure and blood lipid and HbA1c levels.

**Table 4 biomedicines-11-02211-t004:** Univariate predictive model of DKD.

Index	AUC (95% CI)	Specificity	Sensitivity	Cut-Off
FT4	0.57 (0.54, 0.60)	0.44	0.68	15.95
FT3	0.61 (0.58, 0.64)	0.54	0.64	4.85
FT3/FT4	0.65 (0.62, 0.68)	0.41	0.84	0.29

Note: FT4 is for free thyroxine, FT3 is for free triiodothyronine.

## Data Availability

The datasets generated during the current study are available from the corresponding author on reasonable request.

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
