# Peer review of "Predictive Effects of FT3/FT4 on Diabetic Kidney Disease: An Exploratory Study on Hospitalized Euthyroid Patients with T2DM in China"

_biomedicines, 2023, doi:10.3390/biomedicines11082211_

Round 1

Reviewer 1 Report

Study “Predictive effects of FT3/FT4 on Diabetic Kidney Disease: A 2 study on hospitalized euthyroid patients with T2DM in China”, by Zhao and collaborators, is a retrospective study, aimed to establish a regression model for prediction of DKD occurrence in patients with T2DM according to a group of biochemical parameters, including FT3/FT4 ratio.

There are several issues that authors should address, in order to make a manuscript suitable for publishing:

- Exact p values should be given.

- From the table 2 it seems that all the variables were compared by χ2 test – and it is not clear why? It is appropriate only to those that are expresed in proportions. Maybe is better to put: F or χ2.

- Please explain the meaning of “Crude OR” and “Adjust OR”.

- Which method was used to determine the “cut-off” in ROC analysis?

- Please explain a rationale for using all variables in the model.

- Please explain a selection of variables.

- Did authors use a concurrent multivariate predictive model for comparison with the existing (for instance with omitting one of the variables) and what were the results of the comparison?

- Did authors used validation cohort for verification of the model? If they did not, a title should be changed into „Predictive effects of FT3/FT4 on Diabetic Kidney Disease: A 2 study on hospitalized euthyroid patients with T2DM in China; exploratory study“.

Smaller corrections:

- line 76: „or after ingesting 75g of glucose for 2 hours“ – i suppose authors ment „2 hours after ingesting 75g of glucose“

- line 127 - Kolmogorov–Smirnova

- line 245 – „deionyze“ ?

- line 276 – „to the The results“?

Reviewer 2 Report

The authors present a cross-sectional cohort study on the association between thyroid hormone levels and diabetic kidney disease.

The overall rationale of the paper is clear.

The introduction is generally fine. As the primary predictor for DKD is diabetes, the relation between thyroid hormones and diabetes should be presented more clearly. Several papers have reported associations in euthyroid cohorts. (https://pubmed.ncbi.nlm.nih.gov/22751273/; https://pubmed.ncbi.nlm.nih.gov/21104580/, https://pubmed.ncbi.nlm.nih.gov/20447068/, https://pubmed.ncbi.nlm.nih.gov/33125689/, https://pubmed.ncbi.nlm.nih.gov/26049821/)

Methods:

Please clarify the existence of an ethics approval for this study (also in the main text).

Was the study registered in a clinical study register?

How did you determine PBG - by an oral glucose tolerance test for every patient?

Your data allow for bivariate correlation or linear regression analysis between thyroid hormone levels (incl. the ratio) and continuous outcomes of renal function (UACR, eGFR). Please add this analysis.

Results:
Please show, how many patients were diagnosed with DKD based on the individual three diagnostic criteria (eGFR, UACR, biopsy)).

The cut-offs for fT3/fT4 seem to be arbitrarily chosen. Please clarify.

Please use adequate decimals for all parameters, always in accordance to plausible raw data precision (e.g. RR without decimals, BMI with one, WHR and lipids with two decimals...).

Table 1 should provide adjusted p-values for DKD frequency (adjusted for sex, HbA1c, diabetes duration, BP...).

Discussion: Can only be evaluated after major revision.

Citations: Citation style inconsistent.

moderate changes needed

Reviewer 3 Report

The article by X.Zhao et al elucidates the role of the free thyroid hormone ratio in blood serum (FT3/FT4) as a risk factor of the conversion of type 2 diabetes (T2DM) to diabetic kidney disease (DKD). The groups of patients with T2DM were selected properly: large enough, with normal TH levels and randomized by gender, age and body weight parameters. Statistic analysis including logistic regression allowed to develop an univariate (among 3 variables FT3, FT4 and FT3/FT4 the latter appeared the most sensitive) and a multivariate predictive models for the risk of DKD.

All my comments on the article concern the role of deiodinases. It is known that T4 is a pre-hormone produced by thyroid gland. Circulating in blood it is converted to T3 upon entering deiodinase-containing cells and than new T3 returns to circulation. The majority of T3 produced in the body derives from deiodinase 2 (DIO2) activity. The role of deiodinases in FT3/FT4 and the whole TH homeostasis is obvious. Considering this fact it is strange that I found only 1 reference about the possible role of deiodinase 1 (DIO1) inhibition in decreasing FT3/FT4 ratio upon metabolic conditions associated with T2DM (line 44) in Introduction. The description of deiodinases in Discussion section is brief and full of lapses: line 245 “which can deionize” should be deiodinize, line 246 “type 2 has the ability to iodize” should be deiodinize.This part of Discussion should be increased and described properly. It’s a pity that among a large number of measured parameters the activity of DIO2 is out of author’s attention.

In fact the authors themselves consider their study as a preliminary one because for instance the relationship between TSH and DKD remains unclear. May be their later studies include the status of DIO2 as additional parameter into statistical analysis.

Minor editing of English language will improve the quality of the article.
